# Relationship between Oral Health Status and Postoperative Fever among Patients with Lung Cancer Treated by Surgery: A Retrospective Cohort Study

**DOI:** 10.3390/healthcare8040405

**Published:** 2020-10-16

**Authors:** Chieko Itohara, Yuhei Matsuda, Yuka Sukegawa-Takahashi, Shintaro Sukegawa, Yoshihiko Furuki, Takahiro Kanno

**Affiliations:** 1Department of Oral and Maxillofacial Surgery, Shimane University Faculty of Medicine & Oral Care Center, Shimane University Hospital, Izumo, Shimane 693-8501, Japan; ioa2thr@med.shimane-u.ac.jp (C.I.); yuhei@med.shimane-u.ac.jp (Y.M.); 2Department of Oral and Maxillofacial Surgery, Kagawa Prefectural Central Hospital, Takamatsu, Kagawa 760-8557, Japan; yuka611225@gmail.com (Y.S.-T.); s-sukegawa@chp-kagawa.jp (S.S.); furukiy@ma.pikara.ne.jp (Y.F.)

**Keywords:** oral bacteria count, postoperative fever, lung cancer, retrospective cohort study, surgery, perioperative oral management, oral care

## Abstract

A retrospective observational study using an oral bacteria counter was conducted to evaluate the trends in the number of oral bacteria in the perioperative period of lung cancer patients and to verify the relationship between oral health status and postoperative fever. All patients received perioperative oral management (POM) by oral specialists between April 2012 and December 2018 at Kagawa Prefectural Central Hospital, Kagawa, Japan prior to lung cancer surgery. Bacteria counts from the dorsum of the tongue were measured on the day of pre-hospitalization, pre-operation, and post-operation, and background data were also collected retrospectively. In total, 441 consecutive patients were enrolled in the study. Bonferroni’s multiple comparison test showed significantly higher oral bacteria counts at pre-hospitalization compared to pre- and post-operation (*p* < 0.001). Logistic regression analysis showed that body mass index, performance status, number of housemates, number of teeth, and white blood cell count at pre-operation were significantly associated with postoperative fever. The study showed that POM can reduce the level of oral bacterial counts, that the risk of postoperative complications is lower with dentulous patients, and that appropriate POM is essential for prevent of complications. Therefore, POM may play an important role in perioperative management of lung cancer patients.

## 1. Introduction

Cancer treatment by multidisciplinary cooperation is recommended in Japan, and dentists/oral surgeons and dental hygienists play an important role in perioperative oral management (POM) [1]. POM was introduced into the Japanese universal health public insurance system in April 2012, and POM has been developed as one of the supportive cares for the main treatment, mainly by maintaining and managing the oral function before and after the treatment of all cancer patients [1]. Although the consensus of POM methods in Japan has not been established nor consented yet, it is roughly divided into organic oral care, mainly for keeping the oral cavity clean and functional, and oral care for maintaining and improving oral function. It is general practice to receive necessary oral treatment, instruction, and management by a dentist/oral surgeon and dental scaling, professional mechanical tooth cleaning, and dental hygiene instruction and care by a dental hygienist [2]. It has been established that the effects of oral care influence other aspects of health, for example, Yoneyama et al. reported in 1999 that oral care for the elderly protected against aspiration pneumonia [3]. Mori et al. also reported that oral care decreased the incidence of ventilation-associated pneumonia (VAP) in intensive care unit (ICU) patients in a nonrandomized trial, with historical controls using 1666 mechanically ventilated patients admitted to the ICU in 2006 [4]. Since then, there have been increasing reports that oral care is effective in preventing postoperative complications [4]. Recently, it has been reported that oral care is also effective in preventing surgical site infection (SSI) in a multicenter retrospective analysis of 698 patients using analysis of covariance with propensity score matching [5]. In the esophageal region, two reports (a multi-center retrospective study of 280 patients who underwent an esophagectomy and a single-institute and historical cohort study of 341 patients) showed that periodontal disease and non-POM are risk factors for infectious complications after surgery [6,7]. In addition, a nationwide administrative claims database study verified that POM by a dentist and a dental hygienist significantly reduced postoperative complications in patients who underwent cancer surgery [8]. Another large database research also showed that POM in cancer patients was effective in reducing the incidence of pneumonia in hospitals and thereby helped in preventing pneumonia during hospitalization in 2020 [1]. Hence, evidence regarding the effects of POM on patients undergoing surgical treatment including cancer treatment is becoming established.

In lung cancer patients, there have been two reports regarding the effectiveness of POM for prevention of adverse events. Kamiyoshihara et al. reported in their study that two (1.3%) of the 70 patients had postoperative complications, but there were no serious adverse events due to the oral care intervention [9]. Nishino et al. also reported that POM can prevent postoperative pneumonia in lung cancer patients [10]. However, since both papers were reported using a low evidence study design that compared lung cancer patients to those with other cancers (e.g., digestive organ cancer), there is still insufficient evidence for the effect of POM in lung cancer patients. Furthermore, the above-mentioned papers have two shortcomings. First, the effect of POM is based on the premise and hypothesis that aspiration of oral bacteria and infection of oral bacteria will cause surgical site infection, but there is little evidence that demonstrates the cause and effect with the number of oral bacteria. It has been pointed out that the obligate anaerobic Gram-negative bacillus, which is often found in the oral cavity, is one of the causes of VAP development [11], but few studies have shown an association between bacterial counts and postoperative complications.

It is also widely known that body mass index (BMI) has an impact on subsequent outcomes and postoperative complications including postoperative fever in lung cancer surgery patients. Sato et al. reported that surgical intervention is feasible and potentially effective for primary lung cancer but may not achieve positive perioperative and long-term outcomes for patients with a low BMI [12]. Lewis et al. reported in their review that a normal BMI/less weight loss is associated with significant survival improvement [13]. Agostini et al. reported that potentially modifiable risk factors of postoperative pulmonary complications following thoracic surgery include BMI [14]. Im et al. reported that curative resection for non-small cell lung cancer in healthy elderly patients appeared feasible with 10% postoperative pulmonary complications, and lower BMI is a predictor of postoperative pulmonary complications development, which guide treatment decision-making in these patients [15]. On the other hand, Yang et al. investigated the perioperative risk factors of postoperative pulmonary complications after minimally invasive anatomic resection for lung cancer, and they indicated that the incidence of postoperative pulmonary complications was 24.8%, and logistic regression analysis revealed that BMI ≥ 24.0 kg/m^2^ (vs. < 24.0 kg/m^2^: odds ratio of 1.514, 95% confidence interval of 1.057–2.167) was an independent risk factor for postoperative pulmonary complications [16]. Thus, although it is clear that BMI is a risk factor in lung cancer surgery patients, it remains inconclusive whether low or high BMI, or both, is a risk factor. In other words, it was necessary to set up a study design that excluded confounding factors by performing a stratified analysis by BMI.

The gold standard for measuring the number of bacteria in the oral cavity is a method in which saliva, plaque, etc. are sampled, cultured, and then the colony forming units (CFUs) per sample are counted [17]. However, this gold standard method has limitations in terms of cost and labor, and it is difficult to carry out in a study with a large sample size. In recent years, a device has been developed based on a dielectrophoretic impedance measurement (DEPIM) method that can measure oral bacteria objectively, rapidly, easily, and non-invasively. Hamada et al. reported that the objective DEPIM results were in good agreement with the conventional culture method, which has shown the applicability of the DEPIM apparatus for practical and rapid oral bacteria measurement [18]. Currently, the oral bacterial counter has been commercialized and is widely used in clinical and research fields [19]. In clinical research applications, Suzuki et al. reported changes in the number of oral bacteria during the perioperative period, and they suggested that the oral bacteria count is elevated just after surgery, especially if the patient has endotracheal intubation, which may increase the risk of aspiration pneumonia [20]. However, in patients undergoing lung cancer surgery, since the surgical site is the lung itself, it is difficult to accurately evaluate and verify the onset of aspiration pneumonia. Therefore, clinical studies that verify the effects of POM in lung cancer patients may require outcomes that differ from the incidence of aspiration pneumonia. Inai et al. investigated the relationship between postoperative fever and POM in 471 perioperative patients, and reported that oral findings were a risk factor for postoperative fever [21]. In addition, many studies have reported that potentially modifiable risk factors of postoperative pulmonary complications following thoracic surgery include BMI. Therefore, we conducted a retrospective observational study using the oral bacteria counter to evaluate the trends in the number of oral bacteria in the perioperative period of lung cancer patients and to verify the relationship between the oral health status and postoperative fever as a perioperative complication in respect to BMI.

## 2. Materials and Methods

### 2.1. Regular Perioperative Oral Management in Kagawa Prefectural Central Hospital

From April 2012, dentists/oral surgeons and dental hygienists at the Kagawa Prefectural Central Hospital conducted the following procedures on all cancer patients undergoing surgical treatment: oral evaluations (interviews and assessment of teeth, periodontal tissue, mucous membrane, and dentures, etc.) before hospitalization; oral evaluation, oral care, and oral hygiene instruction the day before the operation; and the same POM the day before discharge. When performing POM, the quantity of bacteria was routinely measured, as mentioned above, by an oral bacteria counter. This was performed three times for each patient as an evaluation index and feedback.

### 2.2. Data Sources and Search Strategy

This was a retrospective single-center cohort study to evaluate the risk factors for postoperative fever. In thoracic surgery performed at Kagawa Prefectural Central Hospital, all lung cancer patients who receive radical surgery are supposed to visit an oral care center for POM. Between April 2012 and December 2018 at Kagawa Prefectural Central Hospital (Kagawa, Japan), all patients included in this study received POM after giving informed consent and after measuring the number of oral bacteria three times using an oral bacteria counter in order to prevent perioperative lung cancer complications. All patients underwent an oral and dental examination before lung cancer treatment by the dentists/oral surgeons and dental hygienists, and a flow diagram of the POM is shown in Figure 1. Furthermore, all patients were also advised to receive regular oral and dental care throughout the perioperative period. This study was conducted with the approval of the Medical Ethics Committee of Shimane University (No. 4041) and the Ethics Committee of Kagawa Prefectural Central Hospital (No. 878). Therefore, patients that did not give informed consent for POM were excluded. Furthermore, we also excluded patients where we were unable to measure the oral bacterial count three times (the day they visited the oral care center before hospitalization, the day before the operation, and the day before discharge as a regular POM course).

### 2.3. Study Variables

The following data were collected: patient characteristics (gender, age, BMI, performance status, Brinkman index, number of housemates, type of cancer, cancer stage, and operation time), intraoral findings (number of teeth, denture use, presence of home dentist, white blood cell counts at pre-operation, serum albumin values at pre-operation, estimated duration of hospitalization (days), and duration of hospitalization (days)). BMI was divided into three groups (underweight (BMI < 18.5 kg/m^2^), normal weight (18.5 kg/m^2^ ≤ BMI < 25.0 kg/m^2^), and overweight (25.0 kg/m^2^ ≤ BMI)) using the World Health Organization (WHO) classification and a previous study [22]. Presence of a home dentist was defined as a person who regularly receives a dental examination within the period of one year [23]. For the operation time, variables were extracted from the operation records for the radical resection of lung cancer performed by a single surgical team at Kagawa Prefectural Central Hospital, Department of Thoracic Surgery.

### 2.4. Oral Bacteria Count

Figure 2A shows the bacteria detection apparatus (Panasonic Healthcare, Tokyo, Japan) that was used to measure the bacteria count in the middle dorsal surface of the tongue on the day of pre-hospitalization, pre-operation, and post-operation by experienced hygienists, based on procedures from a previous study [24]. During measurement, the measurement error due to the procedure was minimized to ensure objectivity. A self-administrated mouthwash was carried out using 50 mL of water before the measurement, the collection site was kept constant, the sample was collected using a universal applicator for standardization of sampling, and the examiners were well trained to calibrate and minimize the deviation (Figure 2B). The results of the oral bacteria counts were stratified into the following categories (CFU/mL): <10^6.5^ (level 1); ≥10^6.5^ to <10^7^ (level 2); ≥10^7^ to <10^7.5^ (level 3); ≥10^7.5^ to <10^8^ (level 4); ≥10^8^ to <10^8.5^ (level 5); ≥10^8.5^ to <10^9^ (level 6); and ≥10^9^ (level 7).

### 2.5. Study Outcomes

Based on the Japan clinical oncology group (JCOG) postoperative complication criteria (Clavien–Dindo classification), during the hospital stay, body temperature was measured at least once a day after the operation. The body temperature was measured by axillary temperature. When the maximum body temperature was 37.0 °C or higher, according to the common toxicity criteria of adverse events (CTCAE) v. 4.0, as reported in a previous study, it was classified as a fever and the duration of the postoperative fever was recorded [25].

### 2.6. Statistical Analyses

All statistical analyses were performed using SPSS version 26.0 software (IBM Japan, Tokyo, Japan). The one-way analysis of variance (ANOVA) statistical test was used followed by Bonferroni’s multiple comparison test among each time point. Univariate and multivariate analyses of the risk factors for the duration of fever were conducted using logistic regression analysis (backward selection method) on each BMI group. A *p*-value less than 0.05 was considered statistically significant. Additionally, since a substantial number of patients had missing data, multiple imputation using an ordinal logistic imputation method was utilized, with the assumption that the missing data were missing at random.

## 3. Results

### 3.1. Patient Demographics and Characteristics

In total, 441 consecutive patients (276 males and 165 females) were enrolled in the study, and their characteristics are shown in Table 1. The median age was 71.0 years, and the median BMI was 22.6 kg/m^2^. The performance status (PS) was 0 in 422 patients (95.7%), 1 in 11 patients (2.5%), 2 in 5 patients (1.1%), 3 in 2 patients (0.5%), and 4 in 1 patient (0.2%). The mean Brinkman index was 450.0. The median number of housemates was 1.0. The type of cancer was non-small cell carcinoma in 364 patients (82.5%), small cell carcinoma in 8 patients (1.8%), and other in 69 patients (15.6%). The cancer stage was 1 in 312 patients (70.7%), 2 in 68 patients (15.4%), 3 in 57 patients (12.9%), and 4 in 4 patients (0.9%). The median number of teeth was 22.0. The number of patients with dentures was 198 (44.9%), and 72 patients had a home dentist (16.3%). The median level of oral bacteria count at pre-hospitalization, pre-operation, and post-operation was 5.3, 4.6, and 4.5, respectively. Median white blood cell counts at pre-operation were 6.0 × 10^3^/μL, and mean serum albumin values at pre-operation were 4.2 g/dL. The median estimated duration of hospitalization was 14.0 days. The median duration of hospitalization was 10.0 days. The median operation time was 223 min. There were 367 patients (83.2%) who experienced fever, and the median duration of fever was 2.0 days.

### 3.2. Longitudinal Changes in the Level of Oral Bacterial Counts

Figure 3 shows the results of the one-way ANOVA test, which showed significant differences in the oral bacteria count level between the time points (*p* < 0.001). Bonferroni’s multiple comparison test also showed significant differences between pre-hospitalization and pre-operation (*p* < 0.001), and there was also a significant difference between pre-hospitalization and post-operation (*p* < 0.001). There was no significant difference between pre-operation and post-operation (*p* = 0.242).

### 3.3. Risk Factors for Postoperative Fever Duration by Logistic Regression Analysis

Table 2 summarizes the results of the logistic regression analysis, which investigated potential risk factors of postoperative fever. In the total data, there were significant correlations between postoperative fever and BMI, PS, number of housemates, oral bacteria count at pre-hospitalization, and white blood cell count at pre-operation in univariate analysis, and between postoperative fever and BMI, PS, number of housemates, and white blood cell count at pre-operation in multivariate analysis. In the underweight group, PS, number of teeth, and serum albumin value at pre-operation were significantly associated with postoperative fever in univariate analysis, and PS and number of teeth remained significantly associated with postoperative fever in multivariate analysis. In the normal weight group, age, PS, number of housemates, number of teeth, white blood cell count at pre-operation, and operation time were significantly associated with postoperative fever in univariate analysis, but only number of housemates and white blood cell count at pre-operation were significantly associated with postoperative fever in the multivariate analysis. In the overweight group, PS and number of teeth were significantly associated with postoperative fever in univariate analysis, and they remained significant in the multivariate analysis.

## 4. Discussion

Since the target population of lung cancer included in this study was patients with a sole surgical indication as an initial treatment, most patients with such lung cancer had relatively early-stage cancer. According to the National Comprehensive Cancer Network guidelines, surgical indications for non-small cell lung cancer are approximately up to stage 3 (T3, N1) [26,27]. In addition, in the case of small cell lung cancer, it is considered to be approximately stage 1 (T2, N0) [28,29]. Since the cases in this study did not require induction chemotherapy or radiation therapy, patients at an earlier stage were selected, as the study target chemotherapy population has been shown to alter the oral flora profoundly [30]. Hence, since this study did not include patients undergoing induction chemotherapy, it is considered that reliable data were obtained with little variation in the target population. Therefore, it was a group with low PS and high activity. Lung cancer is diagnosed in about 125,000 people every year in Japan. It tends to be more common in men, and begins to increase rapidly around the age of 60 [31]. Therefore, since the target population of this study has characteristics similar to the lung cancer patient population receiving only a radical resection in Japan, it is considered that the results can be generalized in Japan.

This study was broadly divided into two main findings. First, it has been shown that an initial pre-hospital POM intervention can reduce the level of oral bacterial counts, and that reduced bacterial counts can be maintained by pre-operative POM, and lower levels of oral bacterial counts can also be maintained postoperatively. To the best of our knowledge, this is the first report to objectively study the trend of oral bacterial counts in lung cancer patients. The reason why the number of bacteria in the oral cavity could be kept low may be not only due to the dentists and dental hygienists who performed organic oral care, but also the effect of patient instruction/guidance regarding the need for oral care and management. Many randomized controlled trials have been conducted on organic oral care, and professional tooth cleaning (PTC) and chlorhexidine gluconate swabs have been shown to reduce oral bacteria counts and postoperative complications such as VAP and SSI [32,33]. In Japan, it is not permitted to use the same concentration of chlorhexidine gluconate as were investigated in these randomized controlled trials, but we postulated that physical plaque removal in PTC is just as effective. Regarding instruction of dental hygiene, Nakata et al. indicated that patients would presumably like to avoid postsurgical complications, therefore, if they were educated that oral health care could reduce the risk of adverse events and was associated with general health, their attitude could likely be modified. In short, POM could improve oral health knowledge and positive attitudes of the patients receiving treatments in cancer surgery [34]. Tanda et al. also reported that the effectiveness of POM was verified by oral examinations, hygiene instructions, supragingival scaling, professional mechanical cleaning of the tooth surfaces and/or dentures, and tongue cleaning as intensive oral care at least 2 days prior to lung cancer surgery [35]. In other words, the effect of POM before surgery is likely to be sustained regardless of the presence or absence of the subsequent intervention. From the above, we proposed that basic physical cleaning of the oral cavity and dental hygiene instruction are important tools for reducing oral bacterial counts in lung cancer patients receiving radical resection.

Second, multivariate analysis showed that a lower number of teeth was significantly associated with postoperative fever in the underweight and overweight groups. The total data analysis showed that the factors of BMI, PS, and white blood cell count at pre-operation were significantly associated with postoperative fever, but these factors are well known risk factors for postoperative fever, as shown in previous studies [14,36,37]. In particular, BMI is considered a risk factor for postoperative complications in various surgical sites [38,39,40]. In other areas of surgery, being obese has been noted to be a predictor of postoperative complications in randomized controlled trials [41], and most previous studies have shown that obesity was a greater risk factor than thinness. In addition, there is a report of an association between BMI and occlusal force and masticatory function [42]. Therefore, it was appropriate that we adopted an analytical approach that eliminated confounding factors by performing a stratified analysis based on BMI. Moreover, our results showed that a low white blood cell count is one of the risk factors for perioperative fever, whereas Munro et al. indicated that routine preoperative measurement showed that the platelet count was abnormally low in less than 1.1% of patients, and that platelet count results rarely, if ever, lead to a change in patient management [43]. In addition, since housemates have been reported to be a predictor of successful surgical treatment for morbid obesity, there is a slight suggestion that the presence of housemates may have an effect on the patient’s BMI prior to surgery and secondarily on complication prevention [44].

For the interpretation of the relationship between oral health status (number of teeth) and postoperative fever, we can refer to a similar study by Inai et al. [21], which was designed to determine the risk factors associated with postoperative complications after surgery under general anesthesia according to respiratory function test results and oral conditions. They indicated that the most important risk factor for pneumonia was edentulism. Hence, the conclusion that edentulism is a risk factor for postoperative complications is very similar to the results of our study, in which a low number of teeth was a risk factor for postoperative fever. In addition, notably, the O’Leary plaque control record (an index of the oral bacterial count) was not associated with the presence of postoperative complications in their study. This is the same result as in our study, where the oral bacterial count value was not a risk factor for postoperative fever in the multivariate analysis. A number of reports show that a low number of teeth and poor oral function are risk factors for pneumonia and fever in institutionalized seniors [45,46]. Certainly, the presence of teeth may increase plaque retention factors, but more teeth are better for any patient, because tooth loss has been shown to affect morbidity, such as pneumonia [47]. In summary, the loss of oral function due to tooth loss may increase the risk of postoperative complications such as fever. Therefore, we suggest that while a higher number of teeth and good occlusal function are better for patients undergoing lung cancer surgery, it is important to reduce the oral plaque count through thorough POM. The greater the number of teeth of the lung cancer patients planning for surgical intervention, the more intense POM should be considered in perioperative management. Under- or over-weight patients scheduled for lung cancer surgery should always see a dentist, oral surgeon, and dental hygienist, especially since reducing oral bacterial counts in dentulous patients through POM is a variable and fast-acting intervention compared to dental prosthetic treatment.

Our study had three limitations. First, factors that can affect the results such as surgical procedure, histopathological type, blood loss, and respiratory history were not reported. Second, the middle dorsal surface of the tongue was used for measurement of oral bacteria count with the bacteria detection apparatus, so the site cannot be treated as a representative value of all oral bacteria and is a substitute indirect objective value. However, this potential source of error due to the procedure is calibrated, and, therefore, it is not necessary to consider it. Third, some biases may exist because of the following reasons: the study had a retrospective single center design, multiple dentists/oral surgeons and dental hygienists performed POM, there were differences in cancer severity among patients, and the included patients were highly literate regarding oral hygiene with an understanding of the need for POM. Thus, it is possible that this may have influenced the results that were associated with postoperative fever, and, therefore, a prospective multicenter study should be conducted in the future.

## 5. Conclusions

This study showed that POM can reduce the level of oral bacterial counts, and it was also suggested that BMI is one of the risk factors for postoperative fever after lung cancer surgery. The risk of postoperative complications is lower with dentulous patients, but appropriate POM is essential for prevent of complication. Therefore, POM may play an important role in perioperative management of lung cancer patients.

## Figures and Tables

**Figure 1 healthcare-08-00405-f001:**
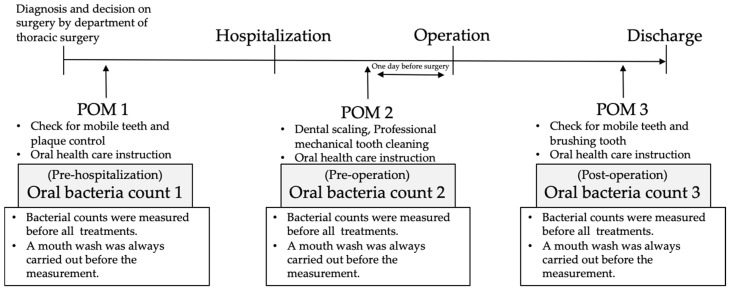
Flow diagram of oral bacteria counts and perioperative oral management (POM).

**Figure 2 healthcare-08-00405-f002:**
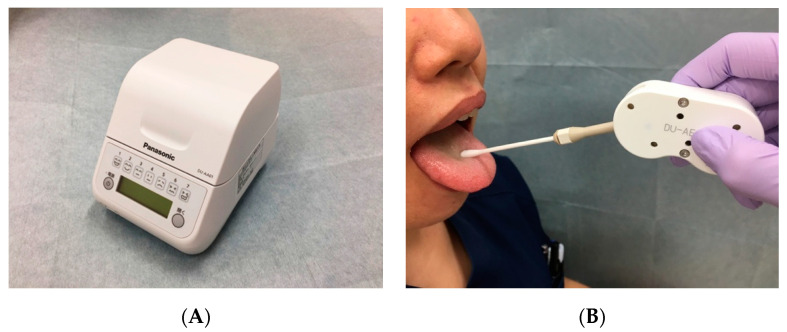
Bacteria detection apparatus (Panasonic Healthcare, Tokyo, Japan). (**A**) The bacteria detection apparatus. (**B**) Sample collection from the tongue.

**Figure 3 healthcare-08-00405-f003:**
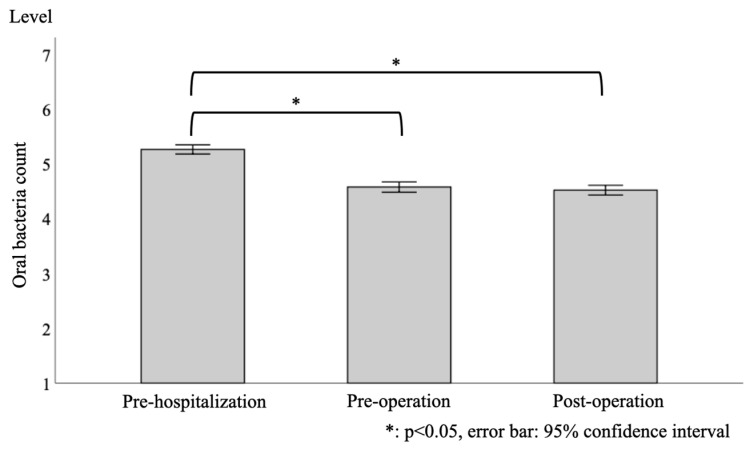
Longitudinal changes in the level of the oral bacterial count (at pre-hospitalization, pre-operation, and post-operation). The one-way ANOVA test showed significant differences.

**Table 1 healthcare-08-00405-t001:** Patient demographics and characteristics.

Characteristics		N (%), Mean (SD) or Median (IQR)			
		All Data (n = 441)	Body Mass Index Categories		
			Underweight (n = 54) BMI < 18.5 kg/m^2^	Normal Weight (n = 286) 18.5 kg/m^2^ ≤ BMI < 25.0 kg/m^2^	Overweight (n = 101) 25.0 kg/m^2^ ≤ BMI
Gender	Male	276 (62.6)	35 (64.8)	171 (59.8)	70 (69.3)
	Female	165 (37.4)	19 (35.2)	115 (40.2)	31 (30.7)
Age (years)		71.0 (64.0–76.0)	73.0 (68.0–78.0)	70.5 (64–75.3)	69.0 (62.5–76.0)
Body mass index (kg/m^2^)		22.6 (20.3–24.8)	17.3 (16.2–17.9)	22.1 (20.6–23.5)	26.6 (25.6–28.1)
Performance status	0	422 (95.7)	49 (90.7)	277 (96.9)	96 (95.0)
	1	11 (2.5)	3 (5.6)	4 (1.4)	4 (4.0)
	2	5 (1.1)	1 (1.9)	4 (1.4)	0 (0)
	3	2 (0.5)	0 (0)	1 (0.3)	1 (1.0)
	4	1 (0.2)	1 (1.9)	0 (0)	0 (0)
Brinkman index		450.0 (0–1000)	380.0 (0–1025.0)	420.0 (0–936.3)	600.0 (0–1157.5)
Housemate (number)		1.0 (1.0–2.0)	1.0 (1.0–2.0)	1.0 (1.0–2.0)	1.0 (1.0–2.0)
Type of lung cancer	Non-small cell carcinoma	364 (82.5)	46 (85.2)	241 (84.3)	78 (77.2)
	Small cell carcinoma	8 (1.8)	0 (0)	2 (0.7)	6 (5.9)
	Other	69 (15.6)	8 (14.8)	43 (15.0)	17 (16.8)
Cancer stage	1	312 (70.7)	35 (64.8)	204 (71.3)	73 (72.3)
	2	68 (15.4)	11 (20.4)	44 (15.4)	13 (12.9)
	3	57 (12.9)	8 (14.8)	36 (12.6)	13 (12.9)
	4	4 (0.9)	0 (0)	2 (0.7)	2 (2.0)
Number of teeth		22.0 (10.0–28.0)	19.5 (4.5–27.0)	23.5 (10.0–28.0)	22.0 (7.5–28.0)
Dentures (yes)		198 (44.9)	29 (53.7)	124 (43.4)	45 (44.6)
Home dentist (yes)		72 (16.3)	35 (64.8)	176 (61.5)	53 (52.5)
Oral bacteria count at pre-hospitalization	(10^6^ CFU/mL)	33.5 (25.0)	31.6 (24.2)	34.6 (25.2)	31.2 (24.9)
	Level	5.3 (0.9)	5.2 (0.9)	5.3 (0.9)	5.3 (0.8)
Oral bacteria count at pre-operation	(10^6^ CFU/mL)	17.9 (16.8)	17.0 (18.2)	18.6 (18.0)	16.1 (11.5)
	Level	4.6 (1.0)	4.5 (1.0)	4.6 (1.1)	4.6 (0.8)
Oral bacteria count at post-operation	(10^6^ CFU/mL)	15.8 (15.0)	17.9 (19.2)	16.2 (14.9)	13.5 (12.3)
	Level	4.5 (1.0)	4.5 (1.1)	4.6 (1.0)	4.5 (0.8)
White blood cell count at pre-operation	10^3^/μL	6.0 (4.9–7.1)	5.6 (4.6–6.8)	6.0 (4.9–7.1)	6.1 (5.0–7.0)
Serum albumin value at pre-operation	g/dL	4.2 (3.9–4.4)	4.1 (3.9–4.4)	4.2 (3.9–4.4)	4.2 (3.9–4.4)
Estimated duration of hospitalization (days)		14.0 (7.0–14.0)	14.0 (7.0–14.0)	14.0 (7.0–14.0)	14.0 (7.0–14.0)
Duration of hospitalization (days)		10.0 (9.0–14.0)	11.0 (8.0–15.0)	10.0 (9.0–14.0)	10.0 (8.0–13.0)
Operation time (minutes)		223.0 (142.0–272.0)	218.5 (153.3–260.8)	226.0 (143.0–273.0)	211.0 (132.5–276.0)
Fever (yes)		367 (83.2)	47 (87.0)	244 (85.3)	76 (75.2)
Duration of fever (days)		2.0 (1.0–4.0)	3.0 (2.0–6.0)	2.5 (1.0–4.0)	2.0 (1.0–3.5)

CFU: colony forming unit, IQR: interquartile range, SD: standard deviation, BMI: body mass index.

**Table 2 healthcare-08-00405-t002:** Prognostic factors affecting the duration of fever after operation by logistic regression analysis.

Univariate Analysis								
Explanatory Variable	All Data		Body Mass Index Categories					
			Underweight BMI < 18.5 kg/m^2^		Normal Weight 18.5 kg/m^2^ ≤ BMI < 25.0 kg/m^2^		Overweight 25.0 kg/m^2^ ≤ BMI	
	Odds Ratio (CI)	*p*-Values	Odds Ratio (CI)	*p*-Values	Odds Ratio (CI)	*p*-Values	Odds Ratio (CI)	*p*-Values
Age					0.96 (0.93–1.00)	0.05		
Body mass index (kg/m^2^)	0.92 (0.85–0.99)	0.02						
PS	0.39 (0.22–0.70)	<0.01	0.38 (0.12–1.10)	0.08	0.49 (0.23–1.06)	0.07	0.09 (0.009–0.90)	0.04
Housemate (number)	0.64 (0.46–0.90)	0.01			0.59 (0.38–0.92)	0.02		
Number of teeth			0.91 (0.82–1.02)	0.10	1.03 (1.00–1.06)	0.09	0.96 (0.91–1.00)	0.06
Oral bacteria count at pre-hospitalization	0.72 (0.53–0.99)	0.04						
White blood cell count at pre-operation	0.83 (0.74–0.94)	<0.01			0.79 (0.69–0.92)	<0.01		
Serum albumin value at pre-operation			0.08 (0.005–1.24)	0.07				
Operation time (minutes)					1.00 (0.999–1.01)	0.10		
Multivariate analysis								
Body mass index (kg/m^2^)	0.92 (0.85–0.99)	0.03						
PS	0.38 (0.21–0.70)	<0.01	0.04 (0.003–0.67)	0.02			0.05 (0.004–0.62)	0.02
Housemate (number)	0.60 (0.42–0.86)	<0.01			0.58 (0.37–0.92)	0.02		
Number of teeth			0.73 (0.53–0.99)	0.05			0.94 (0.89–0.99)	0.03
White blood cell count at pre-operation	0.86 (0.75–0.98)	0.02			0.78 (0.67–0.91)	<0.01		

CI, 95% confidence interval.

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
