# Peer review of "Relationship between Oral Health Status and Postoperative Fever among Patients with Lung Cancer Treated by Surgery: A Retrospective Cohort Study"

_healthcare, 2020, doi:10.3390/healthcare8040405_

Round 1
Reviewer 1 Report
Reviewer comments to the authors:
Thank you for your recent Healthcare submission for review. The present single-center cohort study aimed to retrospectively investigating the relationship between oral health status and postoperative fever among patients with lung cancer. For this study 441 patients were enrolled and all of them have received perioperative oral management (POM) prior to lung surgery. Bacteria counts were measured in the tongue before and after surgery by means of dielectrophoretic impedance measurement (DEPIM). The results indicated that there were significant decrease in oral bacteria counts in the baseline compared to pre and post-operation, and that the reduced level of oral bacteria counts can be maintained post-operatively. Moreover, low number of teeth was associated with postoperative fever. Collectively, the results indicate that POM reduces oral bacteria counts and the risk of postoperative complications.
The aim of this study is interesting and has some merit. The reviewer appreciates that the contents are covering a wide range of information and are written in a way clear enough for the readers to extend their knowledge. It is important to investigate pre-operative oral care in patients that will undergo surgery andI appreciate the efforts made to delve further into a subject matter with a legitimate possibility of adding to our clinical body of knowledge. That being said, I have only minor comments regarding the article as described below.
Please discuss the limitations of the study design such as the biases related to multiple dentists/oral surgeons and dental hygienists performing POM, and the differences in cancer severity among patients.
Please clarify if the persons responsible to count the bacteria were trained and calibrated?
Yours sincerely,
Author Response
Response to Comments/Suggestions from Reviewer 1
Dear Reviewer 1:
We are truly grateful for your critical comments and thoughtful suggestions on our manuscript. We have made careful modifications accordingly. All changes made to the main text are in red. Please find our point-by-point responses to your comments/questions below.
Sincerely,
Prof. Takahiro Kanno, Corresponding Author for the article: healthcare- 962623
Comments and Suggestions for Authors
Please discuss the limitations of the study design such as the biases related to multiple dentists/oral surgeons and dental hygienists performing POM, and the differences in cancer severity among patients.
Response: We would like to thank the reviewer for the comment. As the reviewer pointed out, we have added the limitations of the study design, such as the biases related to multiple dentists/oral surgeons and dental hygienists performing POM. Moreover, we have discussed the differences in cancer severity among patients (Page 11, Lines 316–320).
Please clarify if the persons responsible to count the bacteria were trained and calibrated?
Response: We would like to thank the reviewer for the question. Please note that we have revised this part and added the sentence “standardization of sampling, and the examiners were well trained to calibrate and minimize the deviation” (Page 4, Lines 160–161).
Reviewer 2 Report
The manuscript “Relationship Between Oral Health Status and Postoperative Fever Among Patients with Lung Cancer Treated by Surgery: a Retrospective Cohort Study” presents a retrospective study on the relationship between oral health and postoperative complications in lung cancer surgery not treated with chemo- and radio-therapy. In particular, in addition to other general parameters, the number of teeth present in the mouth and the administration of interventions aimed at improving oral health are taken into consideration.
In the manuscript, the question is original and well defined and the results provide an advance in current knowledge; the results are interpreted appropriately; all conclusions are justified and supported by the results; the article is written in an appropriate way; the data and analyses are presented appropriately; the study is correctly designed and technically sound; the analyses are performed with the highest technical standards; the methods, tools, software, and reagents are described with sufficient details to allow another researcher to reproduce the results; the conclusions are interesting for the readership of the Journal and the paper presumably will attract a wide readership; there is an overall benefit to publishing this work; the English language is appropriate and understandable.
Thera is a typo in line 142: “number of housemates number”.
The only comment that I would like to suggest to the authors is the following: within the limitations of the study it must be clearly specified that the study is retrospective.
For the reasons listed above, my final recommendation is to accept after minor revisions the manuscript.
Best regards
Author Response
Response to Comments/Suggestions from Reviewer 2
Dear Reviewer 2:
We are truly grateful for your critical comments and thoughtful suggestions on our manuscript. We have made careful modifications accordingly. All changes made to the main text are in red. Please find our point-by-point responses to your comments/questions below.
Sincerely,
Prof. Takahiro Kanno, Corresponding Author for the article: healthcare- 962623
Comments and Suggestions for Authors
Thera is a typo in line 142: “number of housemates number”.
Response: We would like to thank the reviewer for the comment. As the reviewer pointed out, we have changed the part “number of housemates number” to “number of housemates.”
The only comment that I would like to suggest to the authors is the following: within the limitations of the study it must be clearly specified that the study is retrospective.
Response: We would like to thank the reviewer for the comment. As the reviewer pointed out, we have clearly specified that the study was retrospective (Page 11, Line 316–321).
Reviewer 3 Report
I read this article.
This article shows importance of preoperative oral management.
There is still insufficient evidence for the effect of POM in lung cancer patients, especially the number of oral bacteria. Furthermore, few studies have shown an association between bacterial counts and postoperative complications. Moreover, this study focused BMI.
This study showed that POM can reduce the level of oral bacterial counts, and that the risk of postoperative complications is lower with dentulous patients, and appropriate POM is essential for prevent of complications. Additionally, this study revealed that BMI is one of the risk factors for postoperative fever after lung cancer surgery.
Weakness of this study is a retrospective single center study. Therefore, in Discussion section, it should be added that prospective multicenter study is required.
Author Response
Response to Comments/Suggestions from Reviewer 3
Dear Reviewer 3:
We are truly grateful for your critical comments and thoughtful suggestions on our manuscript. We have made careful modifications accordingly. All changes made to the main text are in red. Please find our point-by-point responses to your comments/questions below.
Sincerely,
Prof. Takahiro Kanno, Corresponding Author for the article: healthcare- 962623
Comments and Suggestions for Authors
Weakness of this study is a retrospective single center study. Therefore, in Discussion section, it should be added that prospective multicenter study is required.
Response: We would like to thank the reviewer for the comment. As the reviewer pointed out, we have added the part “a prospective multicenter study should be conducted in the future” in the Discussion section (Page 11, Lines 321–322).